# Joint Deployment and Coverage Path Planning for Capsule Airports with Multiple Drones

**Weichang Sun, Zhihao Luo, Kuihua Huang and Jianmai Shi \***

Laboratory for Big Data and Decision, College of Systems Engineering,
National University of Defense Technology, Changsha 410073, China; luozhihao15@nudt.edu.cn (Z.L.)
**\*** Correspondence: jianmaishi@nudt.edu.cn

**Abstract:** Due to the advantages of low cost and high flexibility, drones have been applied to urban surveillance, vegetation monitoring, and other fields with the need for coverage of regions. To expand UAVs' coverage, we designed the Capsule Airport (CA) to recharge and restore drones and provide take-off and landing services. Meanwhile, the combination of drones' coverage path planning (CPP) and the deployment of CAs is a crucial problem with few relevant studies. We propose a solution approach to the CPP problem based on selecting scanning patterns and trapezoidal decomposition. In addition, we construct a 0–1 integer programming model to minimize the cost of the distance between CAs and the scanning missions. Specifically, a solution approach based on greedy and clustering heuristics is designed to solve this problem. Furthermore, we then develop a local-search-based algorithm with the operators of CA location exchange and drone scanning mission exchange to further optimize the solution. Random instances at different sizes are used to validate the performance of proposed algorithms, through which the sensitivity analysis is conducted with some factors. Finally, a case study based on the Maolichong forest park in Changsha, China, is presented to illustrate the application of the proposed method.

**Keywords:** drone; capsule airport; coverage path planning; deployment; heuristic

## 1. Introduction

Recently, the small Unmanned Aerial Vehicle (UAV), also known as drone, has been developed very rapidly and become a popular unmanned equipment. The drone has the characteristics of low cost, high speed, and high mobility, which is an ideal solution for urban surveillance, vegetation monitoring, and other tasks. In some traditional fields, it has become a widespread trend for drones to replace human beings to complete jobs. For example, the drones applied in high-voltage transmission inspections can improve efficiency and accuracy and lower injuries and fatalities [1]. In addition, cleaning drones are capable of fast and efficient disinfection operations. These applications have in common with observing all points of a region of interest (ROI). Since the development of drones, it has become a new research topic to solve the problem using drones.

Solving the coverage path planning (CPP) problem is essential to achieve the applications, which aim to compute a path for drones to cover the whole area. However, the complexity of this problem is proven to be NP-Hard [2]. Currently, some studies project the flight of drones into two dimensions to simplify this problem. Most studies analyze the shape of the area and the flight path of the drone and propose some practical methods to determine a pre-planned route for the drone.

However, due to the limitation of battery capacity, the flight distance and the coverage range of drones are relatively short. This distance limitation is even more unacceptable when faced with scenarios in large areas so that drones are often unable to complete their pre-planned routes. Some common practices include increasing the number of drones or changing batteries frequently. Although these approaches can solve the difficulties to some

extent, they also take up more time and energy and inevitably increase the consumption of human resources. In particular, the power conditions required by drones are also difficult to meet when deployed in field environments or areas with weak electrical infrastructure. These difficulties motivate us to find solutions to extend the range of drones.

In this paper, a capsule airport (CA) is proposed and designed as a platform for integrated drone storage, take-off, landing, solar power generation, charging, and environmental detection. As shown in Figure 1, the CA can hold several small drones inside, and it has a solar power generation function, which can automatically store power to charge the drones. When several CAs are deployed over a large area, these multiple drones can depart separately, return after completing a mission to change batteries, and then quickly execute the next mission. In this way, drones can be applied to a much larger area without the support of human and electrical facilities.

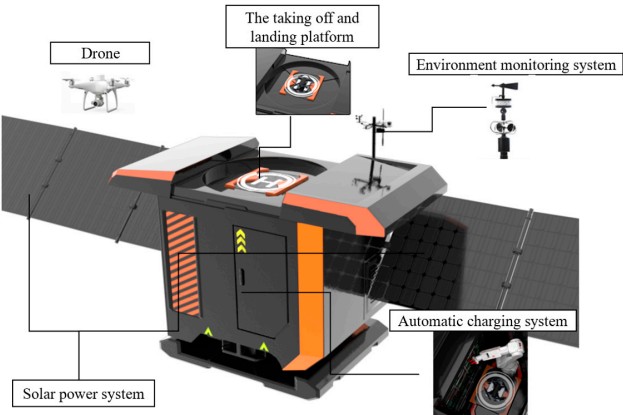

**Figure 1.** Capsule airport and the subsystem components.

The deployment of CA has a significant effect on the route of the drone, and deploying CA in some locations may allow the drones to have shorter flight distances. However, to better utilize CAs, the decision makers must simultaneously optimize the locations for CAs and their drones, a typical location routing problem (LRP) problem in operations research. Thus, this paper studies drones' coverage path planning (CPP) and the deployment of CAs to achieve autonomous and sustainable coverage of a large area by multiple drones at the lowest cost. Compared with traditional research on the CPP problem and LRP, the proposed method is specially designed and efficient to deal with the relationship between the CA deployment and scanning task allocation. In addition, it has advantages in dealing with drone CPP problem with range limitation.

Although the CPP problem and LRP have been widely studied, there are many new challenges to optimizing the combination of the two problems. First, the drone must take off and land in CA, which increases the difficulty in path planning for the drone. Second, the deployment of CAs is relatively complex, so one must consider the additional constraints related to CAs and the assignment of their drones. Therefore, the proposed approach is to model and solve the CA deployment and drones scanning mission assignment problem based on the results of path planning for area coverage. This two-stage approach inherits the general strategy of cooperative multi-drone coverage and is also a simplification of complex constraints.

The problem of drones' coverage path planning and deployment of CAs introduces a new unique idea for drones applied to large-area research and motivates more practical future studies on the practical instances. Our main contributions are as follows:

- A new application mode for drones is designed, which integrates them with the CA to solve CPP problem with the capacity limitation.
- We propose a solution approach to the CPP problem through combining the scanning pattern, optimal sweep direction, and the trapezoidal decomposition method.

- A 0–1 integer programming model for CAs' deployment and drones' scanning mission assignment is established to minimize the overall cost.
- Two heuristic algorithms integrating the ideas of greedy and clustering methods are developed to construct a feasible initial solution in a short time, and a local-search-based algorithm is proposed to improve the quality of the initial solution.
- Both random instances and a practical case are used to illustrate the performance of proposed algorithms, which indicates our methods can efficiently solve the problem.

The paper is structured as follows. Section 2 provides a review of the relevant literature. The problem description and model are presented in Section 3. Section 4 illustrates the heuristic algorithms in detail. The experimental studies are provided in Section 5, including random experiments, sensitivity analysis, and the case study. Finally, Section 6 reports on the conclusion and discusses future research.

## 2. Literature Review

In recent years, coverage path planning techniques for drones have received extensive attention and research, and at the same time combining drone paths with traditional location-routing problems is a research direction worth exploring. The relevant literature is summarized as follows.

### 2.1. UAV Coverage Path Planning

The goal of coverage path planning for UAVs is normally to find the shortest path or shortest completion time to achieve complete coverage of the area or to have all points of interest in the area covered at least once [3,4]. However, in several studies, the objective function is expressed differently. For example, Li et al. [5] establish a model that maximizes coverage and minimizes task time.

According to the difference in the flight path, there are two main types of coverage methods for drones: spiral lines and parallel lines. When we consider that drones take parallel lines, Kuo et al. [6] summarize several common paths and refer to them collectively as back-and-forth paths (BFP), which enable drones to reciprocate in the direction of parallel lines. In [7], BFP is applied to the lawn mower in a farm field or the indoor cleaning robot. Xia et al. [8] compare the total length of the spiral path and BFP for several typical polygons, providing a workable approach to choosing the appreciable method. Compared with the spiral path, the BFP has more effective coverage on irregular regions but results in additional time and energy problems [6]. Therefore, when BFP scans the area, the optimization objective usually includes minimizing the number of turns of the drones. Maza et al. [2] mention that the minimum number of turns occurs when the drone is perpendicular to the minimum width of the polygon. When considering the starting and ending points outside the region, Viridiana et al. [9] improve the method and propose a technique called rotating calipers path planner (RCPP), looking for the minimum width from the point to the convex polygon.

In the above study, we find the general approach to the problem of dealing with convex regions. When it comes to a more complex area that is nonconvex or has an obstacle inside, researchers usually decompose the region into smaller sub-regions to reduce the complexity of regions and facilitate the completion of coverage. This decomposition is called cellular decomposition, and the sub-regions are cellular [10]. Cabreira et al. [11] summarize and discuss the exact and approximate cellular decomposition methods which split the area of interest into sub-areas. Trapezoidal decomposition and boustrophedon decomposition are two exact cellular decomposition methods, and they both use back-and-forth motions to scan all areas exhaustively. Thakar et al. [12] define boustrophedon decomposition as two steps: split and merge. Firstly, the nonconvex polygon is divided into several convex subregions by trapezoidal decomposition, and then the subregions are merged to reduce the excess paths. Compared to trapezoidal decomposition, this method can reduce the number of subregions while guaranteeing a shorter coverage path. In addition to these two methods, the grid-based decomposition method is another common technique. For

example, in maritime search and rescue applications, Cho et al. [13] grid the search area planes with the size of the squares equal to the drone's search width, so the drone must traverse each square to complete full coverage of the area. Galceran and Carreras [7] introduce the morse-based cellular decomposition used for scenes with obstacles or non-polygons scenes. In addition, the grid-based method is applied to the short-distance coverage path of a small cleaning robot. Wang et al. [14] establish a functional relationship between grid terrain and energy consumption using a grid-based method to solve the drone's energy consumption and path planning problem.

As the coverage area expands and the capacity of a single drone is limited, researchers have begun to consider a coordinated search strategy with multiple drones, which can bring the benefits of increased efficiency and reduced losses from partial drone failures [15]. Maza et al. [2] assign one drone to each decomposed convex polygon subregion and explore the optimal scanning direction of the drone. Xia et al. [8] calculated the maximum number of drones required according to the region's total area and the maximum scanning area of the drones and decomposed the region using boustrophedon decomposition. In [16], an upper bound on the minimum number of drones required for full region coverage is calculated. A path planning strategy is designed to consider the sub-regions access frequency constraint. In addition, the cooperative search of multiple drones involves the task assignment of drones, which has been classified and studied by some researchers [17]. An overview of the existing literature on the Coverage Path Planning problem is shown in Table 1.

### 2.2. Location-Routing Problem

For multiple drones and the wide-area coverage problem, positioning capsule airports and determining drones' routes are two complex combinatorial optimization problems to reduce the logistic cost in the process. When we consider the inter-dependency of these two decisions [18], the integrated problem is commonly named the location-routing problem (LRP). Similar to its two subproblems, LRP is shown as an NP-hard problem [19].

There are many studies regarding LRP, especially in the field of vehicle logistics and distribution. In [20], the authors mentioned a classification of LRP problems, commonly based on the number of depots, depots capacity, vehicle capacity, and other factors. Since Nagy and Salhi introduced constraints with capacity on depots and vehicles [21], more researchers have focused on this problem, called capacitated LRP (CLRP). Yu et al. [22] establish the extended LRP model to solve multi-region and multi-depot problems.

One of the approaches to LRP is exact solution methods. Laporte et al. use a branch-and-bound algorithm and progressively refine it by considering special constraints [23–25]. Due to the complexity of LRP, exact solution methods are limited and can only take small-scale instances [21]. Therefore, another methodology of heuristic algorithm is widely used, which has a better adaptability to solve large-scale problems. Tuzun et al. 19 decomposed the LRP problem into two stages and proposed a two-stage forbidden search algorithm. Akpunar et al. [26] propose a hybrid of Adaptive Large Neighborhood Search (ALNS) and Variable Neighborhood Search (VNS) heuristics for solving LRP problems with capacity constraints, and the algorithm performance is tested and evaluated. When considering the multi-allocation LRP constrained by vehicle and hub capacity, the ALNS heuristic algorithm is used and compared with the exact solution methods [27]. Kamyla et al. [28] use the simulated annealing and the artificial algae algorithm to handle LRP with item packaging and other constraints. In [21], clustering-based, hierarchical, and dynamic location-routing methods are also introduced.

The capacity of a depot usually has a lower bound to avoid the waste of resources in most literature cited above [29]. The heuristic algorithm performs well in solving approximate solutions to large-scale problems and also has a shorter calculation time [30]. In addition, most of the researchers focus on traditional vehicle routing and depot location problems, with little attention to the flight routing of UAVs and based location.

The above studies show little literature on coverage path planning for drones when battery capacity constraints are considered. At present, there are few methods for choosing

scanning patterns for drones and decompositions of the concave area. In this paper, we design capsule airports (CA) to provide endurance and other services. We propose a solution approach to the CPP problem, including selecting scanning patterns, optimal sweep direction, and decompositions. Due to the complex interactions between the paths of drones and the location of CAs, the existing studies about LRP have difficulty in solving this problem. Therefore, we build the 0–1 integer programming model and propose heuristic algorithms to optimize the paths of drones and deployment locations of CAs efficiently.

**Table 1.** The main literatures on Coverage Path Planning problem.

| Ref. | Problem Characteristics | Scanning Patterns | | Characteristic of the Area | | Decomposition Method | | |
|---|---|---|---|---|---|---|---|---|
| | | Spiral Lines | Back and Forth | Convex | Concave | Trape-Zoidal | Boustro-Phedon | Grid-Based |
| Kuo et al. (2011) [6] | Robotic cleaner, obstacle avoidance. | √ | √ | | √ | | | |
| Galceran and Carreras (2013) [7] | Coverage Path Planning methods for robots. | | √ | | √ | √ | √ | √ |
| Avellar et al. (2015) [17] | Minimize the number of drones and the task time. | | √ | √ | | | | |
| Otto et al. (2018) [4] | Coverage Path Planning for full area. | √ | √ | | √ | | | |
| Cabreira et al. (2019) [11] | Exact and approximate cellular decomposition. | √ | √ | | √ | √ | √ | √ |
| Wang et al. (2019) [15] | Cooperative coverage of multi-drones. | | √ | | | | | |
| Li et al. (2020) [5] | Minimize task time and maximize coverage, weighted targets, limited battery. | | | | | | | |
| Aggarwal and Kumar (2020) [10] | Exact and adaptive cell decomposition. | | √ | | √ | | | √ |
| Cho et al. (2021) [13] | Minimize the size of decomposed area. | | √ | √ | | | | √ |
| Bezas et al. (2022) [3] | Drone swarm, linear area coverage, point-of-interest detection. | √ | √ | | | | | |
| Vazquez-Carmona et al. (2022) [9] | Region-of-interest disinfection. | | √ | √ | | | | |
| Thakar et al. (2022) [12] | Spray path with mobile manipulator. | √ | √ | | √ | | | |
| Feng and Katupitiya (2022) [16] | Preferential coverage of partial region-of-interest. | | | | | | | √ |
| Xia et al. (2023) [8] | A scanning pattern selection method by calculation. | √ | √ | | √ | √ | √ | |

## 3. Problem Description and Model Formulation

### 3.1. Problem Description

For the covered area, whose boundary range is known, it is assumed in this paper that the altitude of the drones remains constant, regardless of ground buildings and terrain

obstacles. The covered area is an irregular closed polygon, and drones fly within the area in a pre-planned manner, using the camera to search the ground. Therefore, it is a typical coverage path planning problem for drones, and the optimization goal is to achieve the full coverage of the area with the shortest distance or the least time cost. The first step is calculating the sweep width by the geometric relationship between drones and ground and camera parameters. Since different scanning patterns can be used to obtain different shapes and lengths of paths, the second is to determine the suitable scanning pattern. In this regard, researchers have proposed various scanning patterns, such as spiral path and the back-and-forth path (BFP), and the BFP is applied more often.

In addition, the shape of the covered area is also an important factor to consider. For simple convex polygon areas, we need to find the optimal sweep direction when drones take the BFP. Further, drones have the shortest flight distances or the smallest number of turns in this direction. On the other hand, for complex nonconvex polygons or larger areas, it is obvious that the path cannot be constructed directly, and we need to decompose the area. Therefore, choosing a suitable decomposition method is commonly used as trapezoidal decomposition, boustrophedon decomposition, etc. Firstly, we propose a decomposition method based on the shape characteristics of the covered region. Secondly, multiple sub-paths are obtained in the decomposed sub-regions using the coverage path generation method for simple convex polygonal regions. Finally, we connect all the sub-paths to form a general path.

Multiple drones working together can significantly improve efficiency, and the combination of drones and CAs can effectively solve the endurance problem of drones. Since CAs are costly to install and maintain, it is important to calculate the best number of CAs to meet the requirement while saving costs and to have as many drones as possible at each CAs to achieve maximum utilization. The deployment locations of CAs should have good terrain, light, and communication conditions. In this paper, we do not study the effects of these conditions but randomly generate candidate deployment locations to verify the model's and algorithm's applicability.

After determining the general path, this path needs to be assigned to multiple drones so that these drones can scan this at the same time. One method we used was to segment this path to obtain a series of sub-paths. We refer to scanning a sub-path by a drone as a scanning mission. In this way, each drone corresponds to one scanning mission, and we make the round-trip distance between the starting and ending points of each scanning mission and their corresponding CA locations a non-working path. Figure 2 illustrates the positional relationship between the locations of candidate CAs and scanning missions, and it is clear that the selection of CA location directly affects the length of the non-working path. In this case, the travelling cost of the drones consists of two components: one is the cost of covering the path, which is related to the length of the working path, and the other is the length of the non-working path which means the location of CAs should be chosen as close to the mission path as possible. Therefore, this is a typical LRP problem with constraints including the number of drones, the number of CAs, and the number of scanning missions. We assume that a drone is assigned to only one scanning mission, and all scanning missions must be completed. In addition, we assume that the drone takes off from a CA and must return to the original CA after completing the mission. Thus, we can establish the constraint relationship between the CA-drones-scanning mission and represent it with 0–1 integer variables. In Section 3.3, we construct a mathematical model to minimize the sum of non-working paths to find the optimal scheme for the drone scanning mission assignment and the CA deployment, which is similar to the LRP problem and is designed to be solved using a heuristic algorithm due to its complexity.

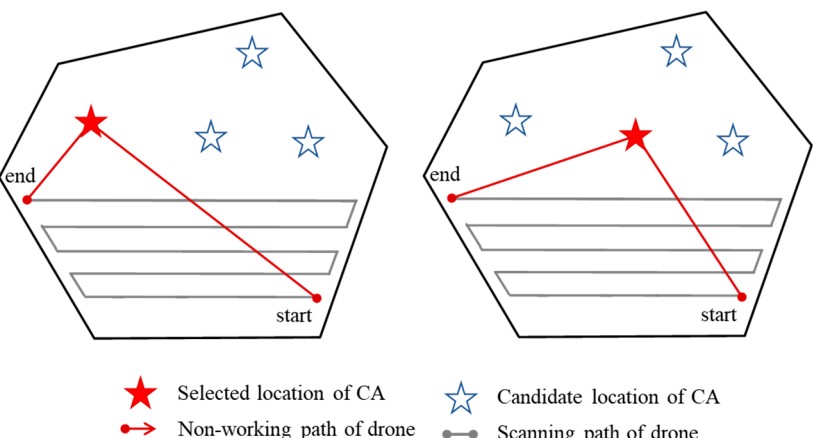

**Figure 2.** The positional relationship between CAs and drone scanning missions.

### *3.2. A Solution Approach to the CPP Problem*

In this section, we propose a solution approach to the CPP problem. When the area is given, we should construct the coverage path according to the shape characteristics of the area. Firstly, we calculate the suitable sweep width between drone paths and propose a method for selecting the suitable scanning pattern for drones. Secondly, we study the optimal sweep direction to obtain the scanning path for drones. Finally, we introduce a decomposition method for the coverage area with a concave shape.

### 3.2.1. Sweep Width and Selection of Scanning Patterns for the Drone

1.  Sweep width

Before the drone performs a coverage scan of the ground area, we need to calculate the sweep width between drone paths. Some researchers have built detection models for drone mission loads [31,32]. The position relationship between the drone and the ground area is shown in Figure 3; it can be seen that the sweep width of the drone is related to the height of the drone and the camera perspective, and the aerial image has certain requirements for the ground resolution. This relationship is shown in the following formula:

$$h = \frac{f \times GSD}{\alpha} \tag{1}$$

where $h$ is the height of drones, $f$ is the focal length of the camera lens, GSD is the cell size, and $\alpha$ is the required resolution of the ground, which is only related to cameras.

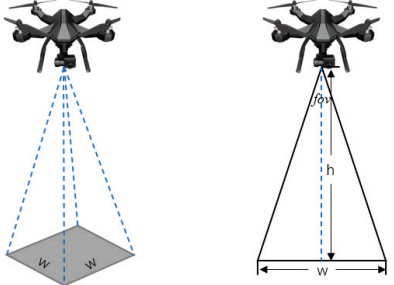

**Figure 3.** Camera footprint of the drone.

According to the height of drone $h$ and the perspective of camera $fov$, the sweep width of the drone is as follows:

$$w = 2htan(\frac{fov}{2}) \tag{2}$$

Considering the image processing requirements, the drone's sweep width must meet a specific overlap rate. As shown in Figure 4, the overlap rate is divided into the heading

overlap rate $\omega_s$ and the side overlap rate $\omega_o$. The heading overlap rate refers to the part of overlapping photos taken before and after when the drone moves forward in a route to take photos, and it is related to the camera shutter time. Most cameras take photos quickly, ensuring the overlapping part between images meets the requirements. The side overlap rate refers to the part of overlapping aerial photos of the drone on two adjacent routes, which is generally not less than 60% according to the research on the aerial photography [33] and some experience from previous literatures [34,35]. Therefore, we can calculate the advanced sweep width $D$ of the drone by using the drone's sweep width $W$ and the side overlap rate $\omega_o$.

$$D = W(1 - \omega_o) \tag{3}$$

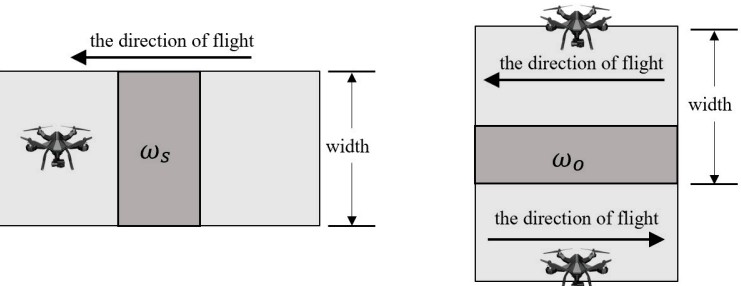

**Figure 4.** Comparison of the heading overlap and the side overlap.

2. Selection of scanning patterns

There are a few specific patterns for coverage paths, such as the lawn mowing and spiral patterns 4. Therefore, in this paper, we consider two typical types of scanning paths based on existing investigations: namely, the spiral path and the back-and-forth path (BFP). A spiral path means that the drone conducts clockwise or counterclockwise spiral scanning with a constant spiral spacing with the center of the coverage area as the spiral center. The BFP means that the drone flies in a straight line, turns at the boundary, and then flies in the opposite direction along a parallel straight line until the entire area is covered.

Though the spiral path and the BFP are commonly used in many kinds of literature [36,37], there are few studies about choosing the more suitable scanning pattern for drones. The flight path length may be significantly different for the same area with varying scanning patterns. For example, for the area shown in Figure 5, a large portion of the spiral path goes beyond the area's boundaries, making it significantly longer than the BFP. It can be assumed that the closer the covered area is to a circle, the shorter the spiral path. Otherwise, the shorter the BFP. In general, the extent to which the covered area is close to a circle is called roundness. In some literature, researchers have studied the definition and calculation methods of roundness [38,39]. Yang introduced a method for calculating the roundness of the projected images of cement particles, which is similar to calculating the roundness of the coverage area [40]. Therefore, we use the method in this paper, and the calculation formula is as follows:

$$e = \frac{4 \times \pi \times S}{L^2} \tag{4}$$

where $e$ is the roundness of the area, $S$ is the area's acreage, and $L$ is the perimeter. When $e$ equals 1, the polygon is a circle, and the smaller $e$ is, the more irregular the polygon is and the greater the difference from a circle.

Therefore, we can choose the more suitable scanning patterns according to the roundness of the coverage area. Xia used a comparison experiment to investigate the relationship between the roundness and the length of the drone's flight path 8. In this study, the two scanning patterns were utilized to cover polygons with different roundness. The result indicated that they have few differences in drone's flight distance when the roundness is 0.86 and show an opposite trend with the change of roundness. In this paper, we take this method and use the roundness of 0.86 as the criterion for selecting the scanning pattern.

When the roundness is more significant than 0.86, we take the spiral path, and conversely, it is better to take the BFP.

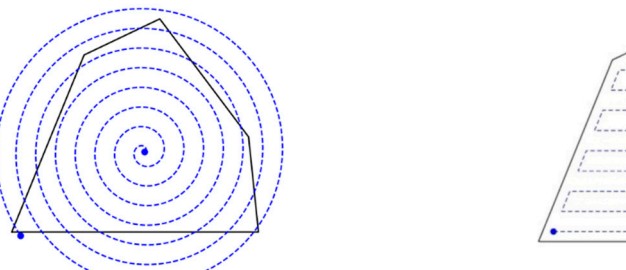

**Figure 5.** Two typical scanning patterns: the spiral path and the BFP.

### 3.2.2. Optimal Sweep Direction

The optimal sweep direction means that the drone has the shortest route when flying in this direction. Researchers usually choose scanning directions based on the shape of the polygon to be covered 17. When the drones take the spiral path, the convexity and concavity of the polygon have almost no effect on the direction selection. However, when the BFP is used, some literature studies the influence of the convexity and concavity of the polygon on the sweep direction [41].

When the BFP is used, drones make a turn at the boundary of the polygon, and this turning process consumes more time and energy, so we believe the number of turns must be reduced. With a determined sweep width, the number of turns of drones in different sweep directions can be calculated. Thus, this problem is translated into choosing the optimal sweep direction of drones [42]. There are various methods for this process 9. Kang [43] use a technique to choose the optimal sweep direction by the minimal width of the polygon. In this paper, we propose a minimum span method to calculate the optimal sweep direction, the shortest path when the drones scan along the direction perpendicular to the minimum span. Li et al. [41] proposed a concept of the span of a convex polygon. First, for any edge of a convex polygon, we calculate the distances between all the vertices on edges separately. Then, the maximum of these distances is recorded as the span of this edge. Finally, the span of each edge is calculated in turn, and the minimum span is found. In this paper, we consider the side corresponding to the minimum span as the long axis of the convex polygon.

In the polygon shown in Figure 6, $d_1, d_2, d_3, d_4, d_5$ are the span of each side of the polygon, where $d_5$ is the minimum span, so the drone makes the least number of turns when scanning along the side $V_1V_2$ that is perpendicular to $d_5$.

### 3.2.3. Decomposition for the Polygons

Considering a given irregular polygon region, the BFP is only adopted when the polygon is convex. Therefore, the convexity and concavity of the polygon need to be determined first since they will have an impact on the planning of the coverage path. Then, the concave polygon should be decomposed into a set of convex sub-regions.

For a polygon $P$, suppose there are $n$ vertices, and the vertex point set is $V = \{v_1, v_2, v_3, \cdots, v_n\}$. Vector fork multiplication can determine the positional relationship between two vectors in space, so we use the vector fork multiplication method to determine the convexity of polygons. Suppose $v_i$ is a vertex of polygon $A$, $v_{i-1}$ and $v_{i+1}$ are two adjacent vertices of $v_i$, and the vector fork multiplication $Q_i$ can be expressed as follows:

$$Q_i = v_{i-1}v_i \times v_iv_{i+1} \tag{5}$$

The convexity of polygon $A$ can be determined by the following method: for all vertices $v_i (i \in \{1, 2, \cdots, n\})$, if $Q_i \geq 0$, then the polygon is convex; otherwise, if there exists a vertex $j \in \{1, 2, \cdots, n\}$ and $Q_j < 0$, then it is a concave polygon.

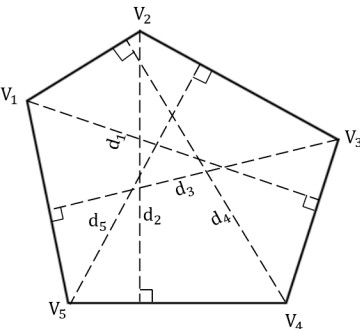

**Figure 6.** The spans of the polygon.

When the region is a nonconvex polygon, we propose a trapezoidal decomposition method that first decomposes the nonconvex polygon into multiple convex polygon subregions and then merges some of the subregions to reduce unnecessary the back-and-forth motions. This decomposition method makes parallel lines along the decomposition direction and divides the polygon into several subregions. The sub-regions after decomposition are combined with the following conditions:

3.  The subregions must be adjacent and have at least one edge that overlaps each other.
4.  The directions of the long axes of the adjacent subregions are parallel to each other.

The trapezoidal decomposition method needs to select the decomposition direction. Similar to the selection of the scanning pattern for the convex polygon area, we use the minimum span method to select the decomposition direction because each convex polygon sub-region after decomposition along this direction usually has a shorter width and fewer turns for drones when they scan this sub-region. It is worth noting that the span concept is for convex polygons. For nonconvex polygons, we can first generate a convex hull, and the minimum span of the resulting convex hull is the minimum span of a nonconvex polygon.

The specific process of trapezoidal decomposition is as follows: first, a Graham Scan algorithm is used to generate concave polygons of convex hulls. As shown in Figure 7a, where the solid line represents the concave polygon, after adding the dashed line, it forms a convex hull; the blue line represents the long axis of the convex hull obtained by the minimum span method, which is also the long axis of the concave polygon. Then, the trapezoidal decomposition method decomposes the polygon along the long axis. Figure 7b shows the result of trapezoidal decomposition, and Figure 7c shows the result of merging some subregions after decomposition. Finally, for each sub-region obtained after decomposition, the drone scans along the optimal sweep direction of each sub-region to generate the BFP of the drone. The result is shown in Figure 7d.

We design the drone area coverage path planning algorithm based on the solution approach to the CPP problem above. First, we input the boundary range of the coverage area and the sweep width of the drone to calculate the roundness of the area by Equation (4). When the roundness is greater than or equal to 0.86, the spiral scanning pattern is selected, and the drone sweep width is spiral pitch. Otherwise, the BFP method is selected. After choosing the scanning pattern, we determine the covered area's convexity by vector fork multiplication method. If it is a concave polygon, we use the Graham Scan algorithm to generate a convex hull of the concave polygon and then use the minimum span method to find the long axis of the convex packet, which is the long axis of the corresponding concave polygon. The drone is made to fly with the BFP along the long axis of each sub-region, and the starting and ending points of the flight path in each sub-region are calculated by traversing the whole sub-region. Finally, we connect all flight paths of different sub-regions in proximity and obtain the drone's total flight path.

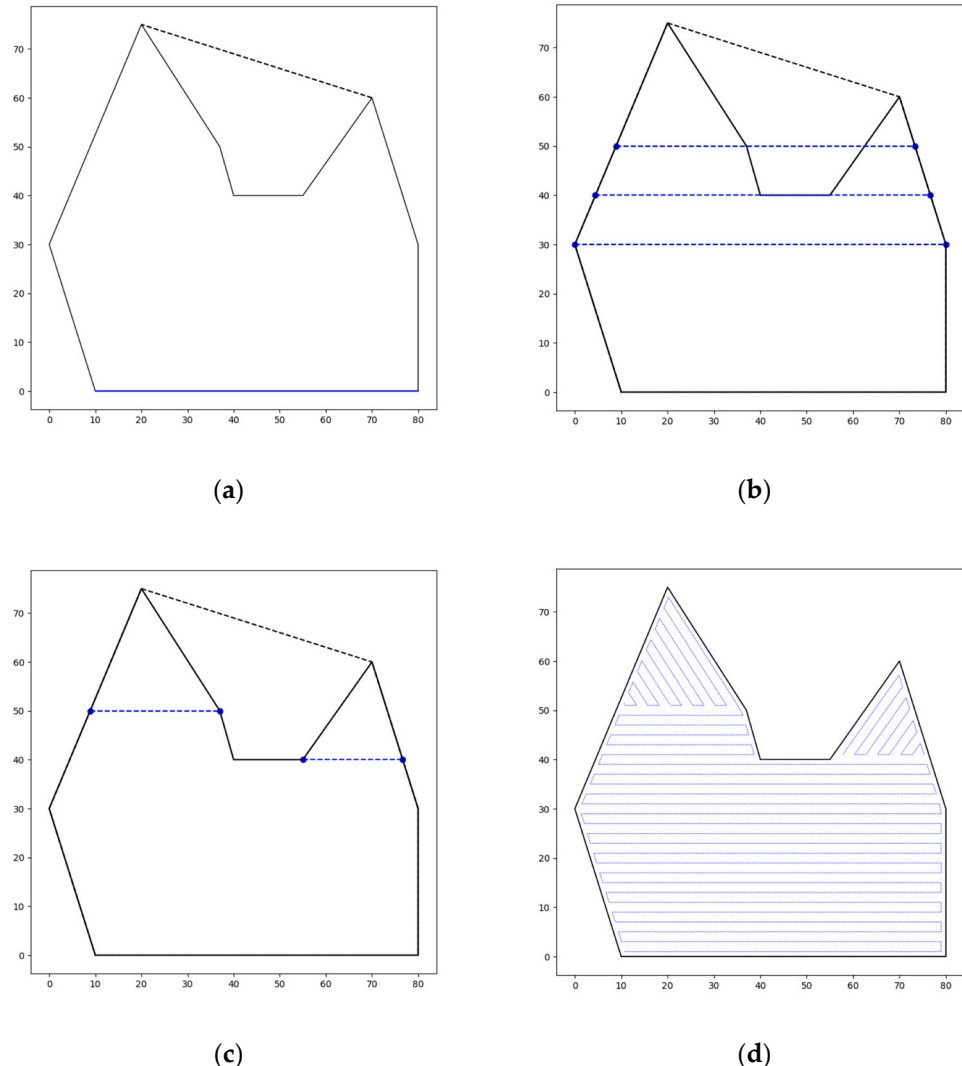

**Figure 7.** Trapezoidal decomposition and the generation of BPF. (**a**) The long axis. (**b**) Trapezoidal decomposition. (**c**) Subregions merging. (**d**) The BFP.

The overall flow of the Algorithm 1 is as follows:

| | Algorithm 1. The drone area coverage path planning algorithm |
|---|---|
| **Input**: | The vertexes $(V_1, V_2, \ldots, V_n)$ of area $A$ |
| **Output**: | The scanning paths $U$ in area $A$ and their length $d$ and the start and end $(sp, ep)$ of the scanning paths |
| 1 | Compute the circumference $L$ and the area $S$ of area $A$; |
| 2 | $C \leftarrow 4 * \pi * S / L^2$; |
| 3 | **if** $C \geq 0.86$ **then** |
| 4 | $U \leftarrow spiral\ scaning\ (A)$; |
| 5 | Compute the start and end $(sp, ep)$ of the scanning paths; |
| 6 | $d \leftarrow distance\ (U)$; |
| 8 | **else** |
| 9 | **end if** |
| 10 | **for** $i \leftarrow 1$ **to** $n - 1$ **do** |
| 11 | $Q_i \leftarrow v_{i-1}v_i \times v_iv_{i+1}$; |
| 12 | **end for** |
| 13 | **if** $\exists\ Q_i < 0$, **then** |

| | |
|---|---|
| 14 | Generate the convex hull of area *A* by the Graham Scan method; |
| 15 | **end if** |
| 16 | Find the direction of decomposition based on the minimum span of the convex hull; |
| 17 | Decompose *A* into multiple convex polygons ($A_i$); |
| 18 | $U \leftarrow$ *spiral scaning* ($A_i$); |
| 19 | Compute the starts and ends ($sp_i$, $ep_i$) of the scanning paths; |
| 20 | $d \leftarrow$ *distance* ($U_i$); |
| 21 | **Return** $U, d, sp_i,\ ep_i$ |

### 3.3. Model Formulation

Using the method above, we obtained the drone paths in each subregion and connected them to a long-distance scanning path. On this basis, the scanning path is segmented to realize the cooperative work of multiple drones. This paper uses the simplest uniform partitioning method to guarantee that each drone is used effectively. First, a series of scanning missions of equal length are obtained and the positions of their start and end points of missions are recorded and numbered. Next, a plurality of candidate deployment locations for use as CAs is generated randomly; these candidate locations can be located inside the region, at the boundary, or outside the region.

The size of available solution space directly affects the optimization results, which is determined by the number of candidate locations. It is obviously that the optimization process is easier but less stable due to the greater influence of randomly generated locations when the number of candidates locations is small. Conversely, it is more likely to find a better deployment solution but with higher computational complexity.

In this paper, the number of candidate locations is set as 3–6 times the number of desired, for the consideration of solution quality and algorithm efficiency. The sensitivity analysis experiment is designed to further analyze this factor in Section 5.2.

For the optimization problem of CAs deployment and the drone scanning mission assignment, we construct a 0–1 integer planning model by minimizing the drones' round-trip distance as the objective function and considering constraints such as the number of CAs and drones and the relationship between drones and scanning mission assignment. The notations used in the model development process are summarized in Table 2.

**Table 2.** Notations and descriptions.

| Sets: | |
|---|---|
| *M* | $= \{1, 2, \cdots, m\}$, the set of the location of candidate CAs; |
| *N* | $= \{1, 2, \cdots, n\}$, the set of segmented trajectories of drones; |
| *V* | $= \{1, 2, \cdots, k\}$, the set of drones; |
| **Parameters:** | |
| $d_{ij}$ | the fight traveling distance between CA *i* and the start and end point of scanning missions *j*, where $i \in M$ and $j \in N$; |
| *P* | the number of CAs; |
| *C* | the maximum number of drones for each CA; |
| **Variables:** | |
| $x_{ijk}$ | 1 if the drone *k* is assigned to CA *i* and takes the scanning mission *j*, 0 otherwise, where $i \in M, j \in N, k \in V$; |
| $y_i$ | 1 if the candidate CA is selected to use, 0 otherwise, where $i \in M$; |

If a certain drone *k* in CA *i* is designated for the scanning mission *j*, and the drone takes off from CA *i*, enters operation from the beginning of that path of scanning mission *j*, completes it at the end, and returns to CA *i* finally, then the distance $d_{ij}$ traveled by the drone *k* on the round trip is as follows:

$$d_{ij} = d_{ij}^q + d_{ij}^d \tag{6}$$

where $d_{ij}^q$ is the flight distance between the start point of mission $j$ and the CA $i$, and $d_{ij}^d$ denotes the flight distance between the end point of mission $j$ and the CA $i$.

The objective function is to minimize the sum of all drones' round-trip distances, so the mathematical model is constructed as follows:

$$\text{Min} \sum_{i \in M} \sum_{j \in N} \sum_{k \in V} d_{ij} x_{ijk} \tag{7}$$

s.t.

$$\sum_{i \in M} y_i = P \tag{8}$$

$$\sum_{i \in M} \sum_{j \in N} x_{ijk} \leq 1, k \in V \tag{9}$$

$$\sum_{i \in M} \sum_{k \in V} x_{ijk} = 1, \forall j \in N \tag{10}$$

$$\sum_{i \in M} \sum_{k \in V} x_{ijk} \leq C y_i, \forall i \in M \tag{11}$$

$$x_{ijk}, y_i \in \{0, 1\}, \forall i \in M, j \in N, k \in V \tag{12}$$

Constraint (8) indicates that the total number of deployed CAs at each candidate location sums to a constant value P. Constraint (9) indicates that each drone is used at most once in a mission. Constraint (10) indicates that each scanning mission must be assigned to one and only one drone. Constraint (11) indicates that the number of drones configured at each CAs does not exceed its maximum capacity. Finally, constraint (12) indicates the range of variable values to be taken.

## 4. Heuristic Algorithm

In this section, two heuristic initial feasible solution construction algorithms and a local-search-based optimization algorithm are designed to solve the CAs setting and drones scanning mission assignment problem. In the first step, the initial feasible solution construction algorithm is used to obtain the initial CAs deployment, and the scanning mission assignment scheme based on the starting and ending points of scanning missions and the locations of candidate CAs. Here, we design two heuristic construction algorithms, the Greedy Match-based Heuristic Algorithm (GMH) and Trajectory Clustering-based Heuristic Algorithm (TCH). We run the two algorithms separately and compare the results to analyze the performance of the algorithms. In the second step, a local-search-based (LS) optimization algorithm is used to improve the initial solution. Here, we design two operators, CA location swapping and the drone scanning mission swapping. These two operators are run sequentially, and the loop iterates to continuously optimize the solution until it satisfies the termination condition. Finally, we output the most satisfactory CAs deployment and drone scanning mission assignment scheme at this time.

### 4.1. Initial Solution Generation Algorithm

In this section, two heuristic algorithms are integrating the ideas of greed or clustering are designed to solve the CAs deployment and drone scanning mission assignment problem—the Greedy Match-based Heuristic (GMH) Algorithm and Trajectory Clustering based Heuristic (TCH) Algorithm. The main solving procedure is shown in Algorithm 2. First, the flight distances between the candidate locations of CAs and the start and end points of drones scanning missions are calculated (line 3). Second, the initial feasible solution is constructed by two heuristic algorithms (line 4). Finally, a local-search-based algorithm is designed to improve the initial solution (line 5).

| **Algorithm 2.** The heuristic algorithm | |
|---|---|
| **Input** | $M, N, C$ |
| **Output** | the optimal CAs deployment and drone assignment scheme $\psi_{optimal}$ |
| **Calculate** | the flight distance $d_{ij}$ between the candidate locations of CAs and the start and end points of scanning missions of drones |
| 1 | Obtain the initial solution $\psi_{initial}$ by Algorithm 3 and Algorithm 4. |
| 2 | Improve $\psi_{initial}$ by Algorithm 5 and get $\psi_{optimal}$ |

## 5. Greedy Match-based Heuristic (GMH) Algorithm

The GMH algorithm is first to obtain the nearest candidate CA–mission pairings for all the scanning missions (line 5), then to select the shortest CA between the candidate CA–mission pairings (line 6). Secondly, we assign the selected CA to the nearest C-1 of the remaining missions (line 7). Thirdly, we select a CA randomly among the remaining candidate CAs and assign it to the closest C missions until all the scanning missions are distributed (lines 11–16). Finally, we obtain the optimal scheme of CAs deployment and drones scanning mission assignment. The algorithm detail is shown in Algorithm 3.

| **Algorithm 3.** Greedy Match-based Heuristic (GMH) Algorithm | |
|---|---|
| **Input**: | $M, N, C, d_{ij}$ |
| **Output**: | $\psi_{initial}^{GMH}$ |
| 1 | **for** $j \leftarrow 1$ **to** $n$ **do** |
| 2 | **for** $i \leftarrow 1$ **to** $m$ **do** |
| 3 | Obtain the nearest candidate CA mission–pairings for all the scanning missions based on $d_{ij}$ |
| 4 | **end for** |
| 5 | **end for** |
| 6 | Select the shortest CA between the candidate –mission pairings |
| 7 | Assign the selected CA to the nearest C-1 of the remaining missions |
| 8 | Update the set of remaining candidate CAs and missions |
| 9 | $m \leftarrow$ the number of remaining candidate CAs |
| 10 | $n \leftarrow$ the number of remaining missions |
| 11 | **While** n > 0 |
| 12 | Select a CA randomly among the remaining candidate CAs |
| 13 | assign the selected CA with the closest C missions |
| 14 | Update the set of remaining candidate CAs and missions; |
| 15 | $m \leftarrow$ the number of remaining candidate CAs; |
| 16 | $n \leftarrow$ the number of remaining missions; |
| 17 | **end while** |
| 18 | Obtain $\psi_{initial}^{GMH}$ by the selected CAs and the assigned missions. |
| 19 | **Return** $\psi_{initial}^{GMH}$ |

## 6. Trajectory Clustering-based Heuristic (TCH) Algorithm

The paths of scanning missions are presented as trajectories here. The trajectory Clustering-based Heuristic Algorithm (TCH) in Algorithm 3 is to use the candidate CAs as clustering centers to find the closest C trajectories to them and keep repeating the process. Firstly, obtain the nearest C of trajectories for all candidate CAs (line 6) and calculate the sum of the distance of trajectories for all candidate CAs (line 9). Secondly, select the CA with the shortest sum of distance, and the corresponding C trajectories are assigned to it (line 10). Thirdly, update the set of remaining candidate CAs and trajectories (lines 9–11). Finally, repeat the process above until all the trajectories are distributed. Finally, we can obtain the optimal scheme of CAs deployment and drone scanning mission assignment. The algorithm detail is shown in Algorithm 4.

**Algorithm 4.** Trajectory Clustering-based Heuristic (TCH) Algorithm

| | |
|---|---|
| **Input**: | $M, N, C, d_{ij}$ |
| **Output**: | $\psi_{initial}^{TCH}$ |
| 1 | **While** $n > 0$ |
| 2 | **for** $i \leftarrow 1$ **to** $m$ **do** |
| 3 | **for** $j \leftarrow 1$ **to** $n$ **do** |
| 4 | Obtain the nearest C of trajectories for all candidate CAs |
| 5 | **end for** |
| 6 | **end for** |
| 7 | Calculate the sum of the distance of trajectories for all candidate CAs |
| 8 | Select the CA with the shortest sum of distance and the corresponding C trajectories are assigned to it |
| 9 | Update the set of remaining candidate CAs and trajectories |
| 10 | $m \leftarrow$ the number of remaining candidate CAs |
| 11 | $n \leftarrow$ the number of remaining trajectories |
| 12 | **end while** |
| 13 | Obtain $\psi_{initial}^{TCH}$ by the selected CAs and the assigned trajectories |
| 14 | **Return** $\psi_{initial}^{TCH}$ |

*4.2. Local Search-Based Optimization Algorithm*

After the initial feasible solution is obtained, the non-working distance between CAs and the drone scanning missions is often largely due to the limitations of the initial solution construction algorithm, which can increase the power consumption of drones. More seriously, the excessively long non-working path can prevent the drone from completing the mission before it must return. Therefore, it is important to optimize the initial solution. In this paper, two domain Local-Search operators (LS) are designed to determine the optimal CAs deployment and drone scanning mission assignment scheme.

7.    CAs position exchange operator

Deploying CAs and assigning drones at the corresponding locations based on feasible solutions can achieve coverage of all trajectories. However, there are cases when other CAs are closer to the assigned trajectories. Therefore, we design the CAs position exchange operator, which refers to randomly selecting one of the selected CAs from the initial solution, closing it, and then randomly selecting one of the unselected sets of candidate CAs to replace the closed one. Then, we assign each trajectory in the selected trajectory set to the closest CA in the selected CAs set. Suppose the number of trajectories assigned to the CA reaches the maximum number. In that case, the trajectories are assigned to the closest of the remaining CAs until all assignments are completed, at which point the new solution is obtained. As shown in Figure 8, CA 11 is closed and replaced with CA 7. In this way, the drones' scanning trajectories can potentially be assigned to closer CAs.

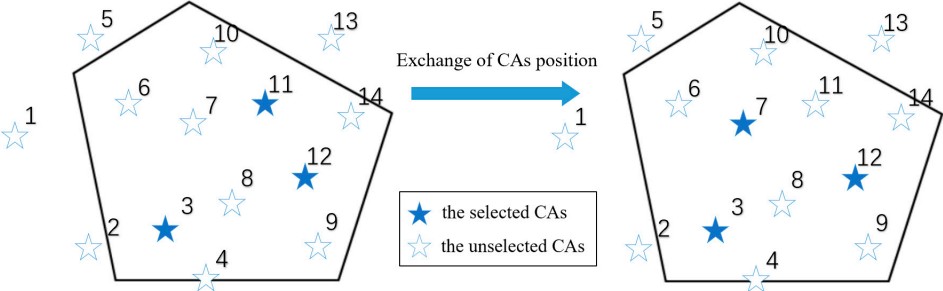

**Figure 8.** Schematic diagram of CAs position exchange operator.

8.    Scanning mission exchange operator

The scanning mission exchange is performed after the CAs position exchange. We randomly choose two CAs from the CAs selected for deployment and exchange their

corresponding scanning missions to obtain a new solution. As shown in Figure 9, CA 1 and CA 2 are selected, and their assigned scanning tasks are exchanged. From the results, the total distance of the drones' non-working path is represented by the dashed line, and its length is significantly reduced.

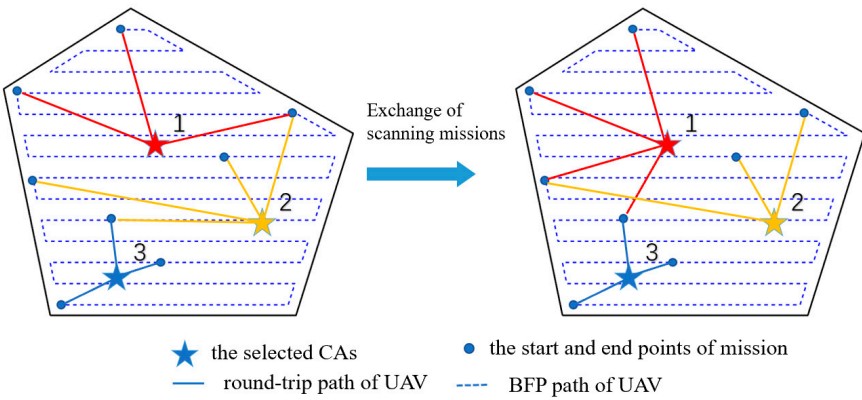

**Figure 9.** Schematic diagram of scanning mission exchange operator.

In the algorithm, the local search is to update the solution by integrating CAs position exchange operator and scanning mission exchange operator. The termination condition is designed to reach the maximum number of iterations. To reduce the cost of time, if the solution did not improve for 50 iterations, we suppose the solution would have been optimal. The local-search-based algorithm is shown as Algorithm 5. With the above heuristic algorithm, the satisfactory solution and the objective function value are obtained: the CAs deployment and drone scanning mission assignment scheme and the total non-working distance of drones.

---

**Algorithm 5.** The local-search-based optimal algorithm.

| | |
|---|---|
| **Input**: | $\psi_{initial}$, $iter_{max}$ |
| **Output**: | $\psi_{optimal}$ |
| 1 | Initialization: $\psi_{optimal} = \psi_{initial}$, $flag = 1$, $iter = 1$ |
| 2 | **while** $iter < iter_{max}$ **do** |
| 3 |   Improve the solution by the CAs position exchange operator and obtain $\psi'$ |
| 4 |   **if** $\psi' < \psi_{optimal}$ **then** $\psi_{optimal} = \psi'$ |
| 5 |   **end if** |
| 6 |   Improve the solution by the scanning mission exchange operator and obtain $\psi''$ |
| 7 |   **if** $\psi'' < \psi_{optimal}$ **then** $\psi_{optimal} = \psi''$ |
| 8 |   **else** $flag = flag + 1$ |
| 9 |   $iter = iter + 1$ |
| 10 |   **end if** |
| 11 |   **if** flag > 50 and **then** |
| 12 |     **Return** $\psi_{optimal}$ |
| 13 |   **end if** |
| 14 | **end while** |
| 15 | **Return** $\psi_{optimal}$ |

---

## 5. Experiments and Results

In this section, we analyze the performance of the proposed GMH-LS and TCH-LS in terms of time and solution quality, and all experiments are run on python 3.7. Randomly generated instances and practical cases demonstrate the effectiveness of the proposed algorithm. The experiments based on randomly generated instances are carried out to explore the performance of the algorithms at different scales. In addition, we further validate the algorithms with a practical case of Maolichong Forest Park in Changsha, China.

In the experiments, we refer to the performance parameters of the DJI "Spirit 4 RTK" drone. The camera image sensor adopts 1-inch CMOS, the lens angle of view is 84°, the focal length is 8.8 mm, the maximum photo resolution is 5472 × 3468, and the ground sampling distance GSD is (H/36.5) cm/pixel. In 1:500 aerial topographic map mapping, the ground resolution is required to be 4–5 cm/pixel. The calculated flight height of the drone is between 146.0–182.5 m, so the flight height is set at 160 m. According to equation 2 and equation 3, the sweep width of the drone is about 200 m. In addition, the maximum flight range is 25 km, and the power consumption efficiency is 1.2:1 when the camera is on and off. Therefore, the working path is 15 km and the non-working path is 7 km when the flight speed is maintained unchanged.

### 5.1. Random Experiment

5.1.1. Experiment Design

As shown in Figure 10, we consider different numbers of CA, including 3, 4, and 5. Accordingly, 9, 20, and 25 candidate locations of CAs are set. Three polygon regions of different shapes are randomly generated under each scale, with nine random instances in total. The candidate locations of CAs are randomly generated, and each CA is equipped with a maximum of four drones. Further, the scanning path of drones obtained by the CPP method in Section 3 is divided into several sub-paths in each randomly generated experiment. The number of sub-paths does not exceed the maximum number of drones owned by CAs, and each sub-path does not exceed 15 km in length.

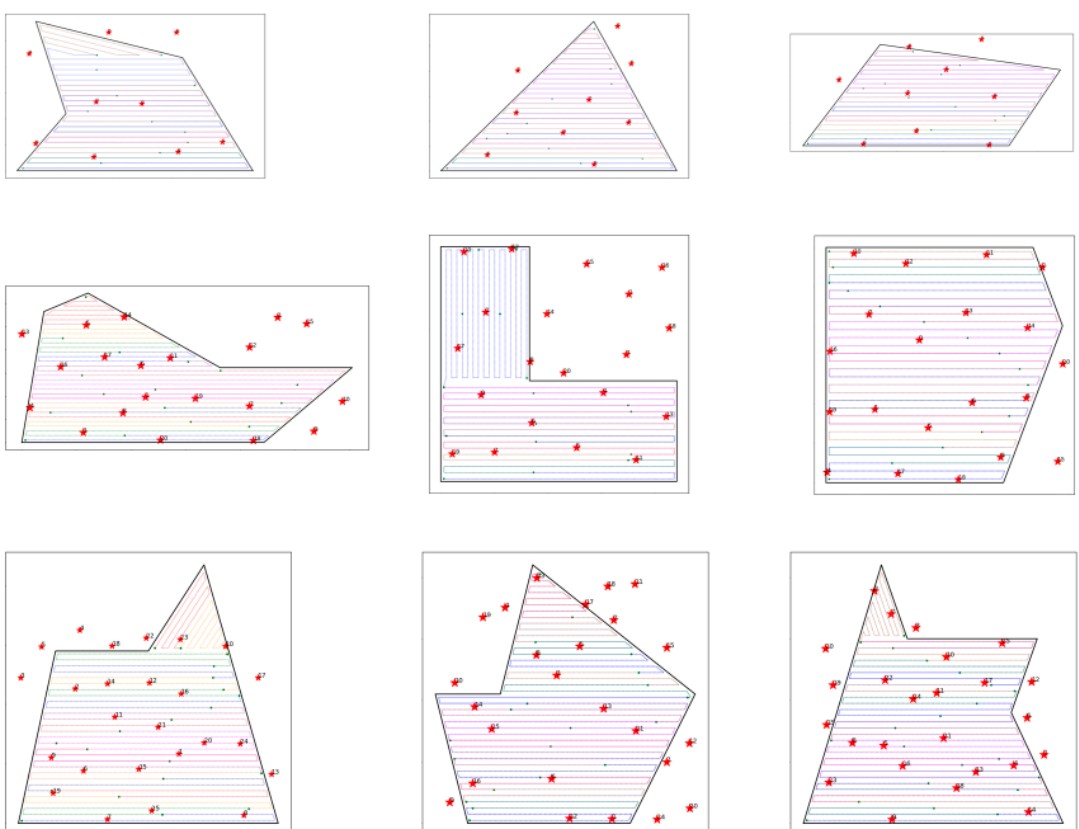

**Figure 10.** Scanning paths of drones and candidate locations of CAs in random instances.

According to the initial feasible solution generation algorithm and the local search optimization algorithm mentioned in Section 4, there are four solution algorithms. We compare the results and solution times obtained by these four algorithms and analyze the algorithms' characteristics during the experiments. The results of the four solution methods are as follows:

1. The result generated by the GMH algorithm and not optimized;
2. The result generated by the TCH algorithm and not optimized;
3. The result generated by the GMH algorithm and further optimized through local search;
4. The result generated by the TCH algorithm and further optimized through local search.

In the local search process, considering the size and solving efficiency, the maximum number of iterations of CA position exchange and scanning mission exchange is set as 1000. If the result is not improved 50 consecutive times, it will be considered as the optimal solution and exit the iteration process.

### 5.1.2. Experiment Results

The computational results for nine random instances with the algorithms of GMH and GMH + LS, TCH, and TCH + LS are presented in Tables 3 and 4. The first column is the number of instances. In the second column, (*m-M-n*) refers to the number of selected CAs, the number of candidate CAs, and the scanning missions of drones. The algorithms of GMH, TCH, GMH + LS, and TCH + LS have been run 10 times. Their best, worst, and average round-trip distances of 10 runs and the average cost time are displayed. To compare the improvement of the local-search-based optimization algorithm on the two initial solutions, we record the improvement of the average round-trip distance of GMH + LS and TCH + LS relative to GMH and TCH.

**Table 3.** Results of nine random instances with the algorithms of GMH and GMH + LS.

| Num | Size (*m-M-n*) | GMH | | | | GMH + LS | | | | |
|---|---|---|---|---|---|---|---|---|---|---|
| | | Best (km) | Worst (km) | Ave (km) | Time (s) | Best (km) | Worst (km) | Ave (km) | Time (s) | Impro (%) |
| 1 | 3-9-11 | 37.65 | 60.63 | 49.56 | $1 \times 10^{-4}$ | 34.25 | 34.25 | 34.25 | 0.053 | 30.89% |
| 2 | 3-9-11 | 40.98 | 62.25 | 50.31 | $1 \times 10^{-4}$ | 37.76 | 38.87 | 38.09 | 0.046 | 27.18% |
| 3 | 3-9-11 | 35.93 | 64.00 | 50.54 | $1 \times 10^{-4}$ | 34.85 | 34.85 | 34.85 | 0.052 | 31.05% |
| 4 | 4-20-16 | 81.58 | 113.25 | 95.16 | $1 \times 10^{-4}$ | 69.34 | 70.85 | 69.75 | 0.054 | 26.70% |
| 5 | 4-20-15 | 72.21 | 144.03 | 101.31 | $1 \times 10^{-4}$ | 54.37 | 56.43 | 55.28 | 0.079 | 45.44% |
| 6 | 4-20-15 | 50.88 | 98.96 | 71.83 | $1 \times 10^{-4}$ | 34.94 | 41.28 | 35.81 | 0.064 | 50.14% |
| 7 | 5-25-19 | 85.09 | 119.19 | 100.75 | $1 \times 10^{-4}$ | 65.97 | 68.81 | 67.33 | 0.084 | 33.17% |
| 8 | 5-25-19 | 104.88 | 133.80 | 119.54 | $1 \times 10^{-4}$ | 77.02 | 80.55 | 78.25 | 0.071 | 34.54% |
| 9 | 5-25-20 | 99.06 | 176.15 | 120.74 | $2 \times 10^{-4}$ | 75.99 | 81.99 | 77.36 | 0.088 | 35.93% |

**Table 4.** Results of 9 random instances with the algorithms of TCH and TCH + LS.

| Num | Size (*m-M-n*) | TCH | | | | TCH + LS | | | | |
|---|---|---|---|---|---|---|---|---|---|---|
| | | Best (km) | Worst (km) | Ave (km) | Time (s) | Best (km) | Worst (km) | Ave (km) | Time (s) | Impro (%) |
| 1 | 3-9-11 | 37.65 | 37.65 | 37.65 | $1 \times 10^{-4}$ | 34.25 | 34.25 | 34.25 | 0.046 | 9.03% |
| 2 | 3-9-11 | 32.25 | 38.25 | 3.825 | $1 \times 10^{-4}$ | 37.76 | 38.29 | 37.87 | 0.043 | 1.11% |
| 3 | 3-9-11 | 43.88 | 43.88 | 43.88 | $1 \times 10^{-4}$ | 34.85 | 38.08 | 35.17 | 0.040 | 19.84% |
| 4 | 4-20-16 | 72.50 | 72.50 | 72.50 | $1 \times 10^{-3}$ | 69.34 | 69.34 | 69.34 | 0.057 | 4.36% |
| 5 | 4-20-15 | 57.74 | 57.74 | 57.74 | $3 \times 10^{-4}$ | 54.37 | 57.74 | 56.30 | 0.060 | 2.49% |
| 6 | 4-20-15 | 37.73 | 37.73 | 37.73 | $3 \times 10^{-4}$ | 34.94 | 34.94 | 34.94 | 0.081 | 7.39% |
| 7 | 5-25-19 | 67.08 | 67.08 | 67.08 | $1 \times 10^{-4}$ | 65.45 | 67.08 | 66.10 | 0.066 | 1.46% |
| 8 | 5-25-19 | 81.64 | 81.64 | 81.64 | $7 \times 10^{-4}$ | 77.02 | 81.59 | 79.92 | 0.080 | 2.10% |
| 9 | 5-25-20 | 77.12 | 77.12 | 77.12 | $8 \times 10^{-4}$ | 77.12 | 77.12 | 77.12 | 0.052 | 0.00% |

We can see that the TCH algorithm can obtain a better initial solution than the GMH algorithm, and this solution's value is always close to the optimal solution. The two algorithms GMH + LS and TCH + LS can obtain a very short round-trip distance after many

optimization times. It shows that the two algorithms can overcome the local optimal and find optimal solutions. However, due to the differences in initial solutions, it is obvious that the GMH + LS has a more significant improvement on the initial solution, from 30% to 50%. In contrast, the TCH + LS algorithm has a small improvement on the initial solution, mostly within 10%.

As for the calculation time, the two initial solution generation algorithms (GMH and TCH) can complete the calculation efficiently, and the time is less than 0.0001–0.001 s. When local-search-based algorithms are used to optimize the solution, the calculation time of the GMH + LS algorithm and the TCH + LS algorithm is also short, ranging from 0.04 s to 0.09 s, which indicates that these two methods have high efficiency and can complete the convergence quickly. With the increasing number of drones and CAs, the time of the TCH + LS algorithm is about 0.02–0.03 s shorter than that of GMH + LS. Therefore, it is acceptable and does not affect the solution.

### 5.2. Sensitivity Analysis

The sensitivity test is conducted on one of the random experiments in Figure 10 with two critical factors: the number and distribution density of candidate locations of CA. Both factors directly impact the selection of CAs' deployment locations. In this experiment, the number of selected CAs is 4, while the number of candidate locations includes 16, 20, and 24. The distribution density of candidate locations of CA is measured by the ratio of the average distance to the area in this paper, which is defined by the following formula:

$$\rho = \frac{\frac{1}{M}\sum_i^M \bar{d}_i}{S} \tag{13}$$

where $\rho$ is the distribution density of candidate locations of CA, $M$ is the number of locations, $S$ is the distribution area, $\bar{d}_i$ represents the average distance between the candidate locations $i$ and the others.

By randomly generating different candidate locations of CA, the distribution density is varied from 0.045 to 0.080, and the result of comparison experiments are presented in Figure 11. As the boxplot is shown, the flight distance of drones decreases at all the values of distribution density when the number of candidate locations are 16, 20, and 24. It can be inferred that when the candidate locations increase, more nearby locations are available to choose from, which can save the overall round-trip distance. Furthermore, when the distribution density is large than 0.060, the flight distance rises slightly. In this situation, due to the extension of the average distance between candidate locations, CAs are deployed more dispersed, which increases the length of non-working routes for multi-drone collaborative scanning missions.

### 5.3. Case Study

#### 5.3.1. Case Description

A case is built based on the part of Maolichong Forest Park in Changsha, China. The area is about 8 km long from east to west and 7 km wide from north to south, giving it the right shape and size. However, the park has an undulating terrain; trees mainly cover the surface. Therefore, it is difficult to monitor by human means, but drones can be effective. Therefore, CAs and drones can be deployed in this area. All data in this case experiment are from the open-source information of Amap.

We smooth the irregular boundary of the forest park and then outline the polygon on this basis, as shown in Figure 12. Then, the actual position of each vertex of the polygon is obtained by extracting the latitude and longitude coordinates. We set up a plane rectangular coordinate system and transformed the latitude and longitude of the polygon vertices into x and y coordinates.

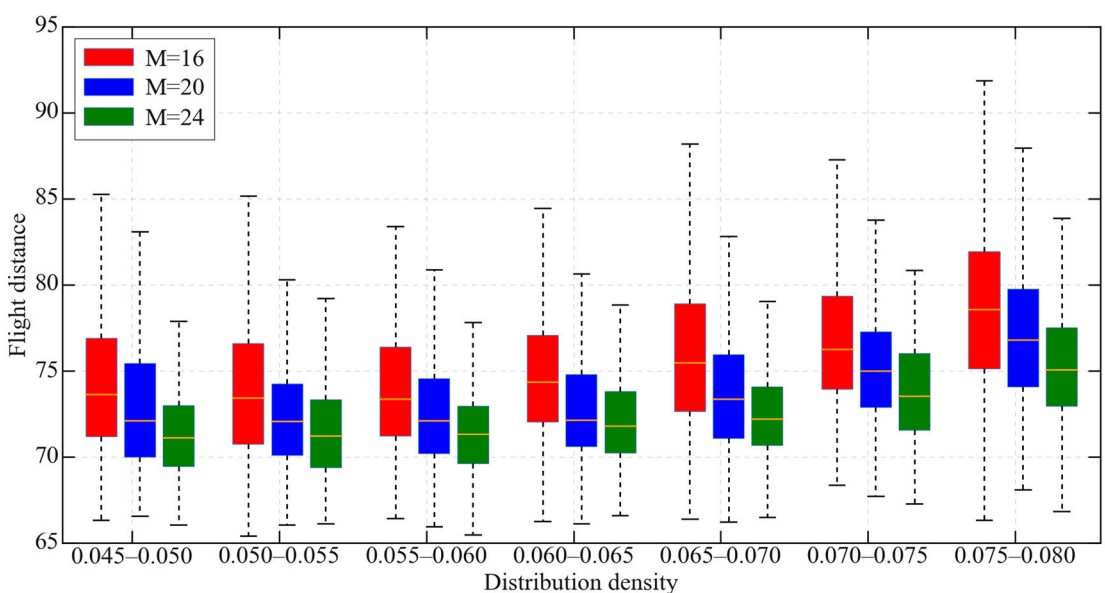

**Figure 11.** Computational results under different distribution densities and number of candidate CAs.

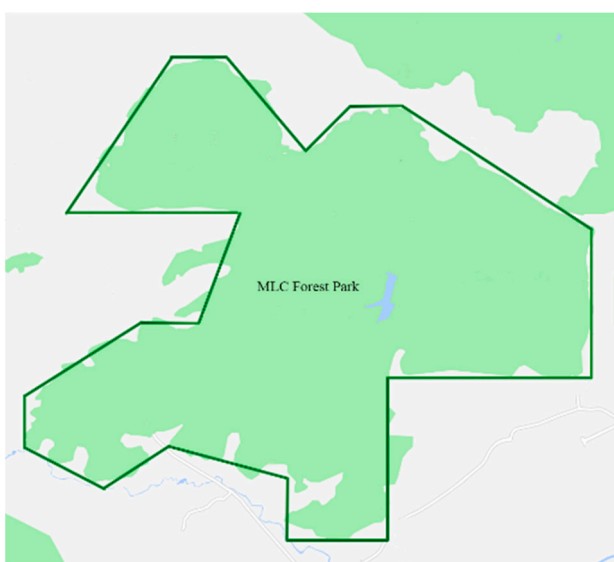

**Figure 12.** The map and abstract polygon of Maolichong (MLC) Forest Park in Changsha.

Similarly, we use the CPP method in Section 3: first, we calculate that the perimeter of the polygon is 30.22 km and the area is 45.63 km². The roundness is calculated by Equation (4) to be 0.43 < 0.86, so the BFP method should be adopted.

In addition, the polygon is an irregular concave polygon, so it can be processed using our proposed trapezoidal decomposition method. The blue dashed line in Figure 13a indicates the optimal decomposition direction and the optimal scanning direction of the drone, and Figure 13b shows the obtained optimal drone scanning path. The total length of this path was calculated to be 152.46 km. Since the scanning distance of each drone is about 15 km, the path is divided into 11 segments; that is, 11 scanning missions, and each mission corresponds to one drone. As shown in Figure 14, the coordinates of the starting and ending points of each segment of the trajectory are calculated and marked with green dots, and a different color represents each segment of the trajectory. Since each CA is equipped with up to four drones, the number of CAs is set to three to reduce CA construction and maintenance costs and improve utilization. Considering the need for multiple alternatives, nine candidate CA deployment locations are randomly generated. In

Figure 14, the numbers 1-9 indicate the serial numbers of the nine candidate CAs, and the red pentagrams indicate their positions.

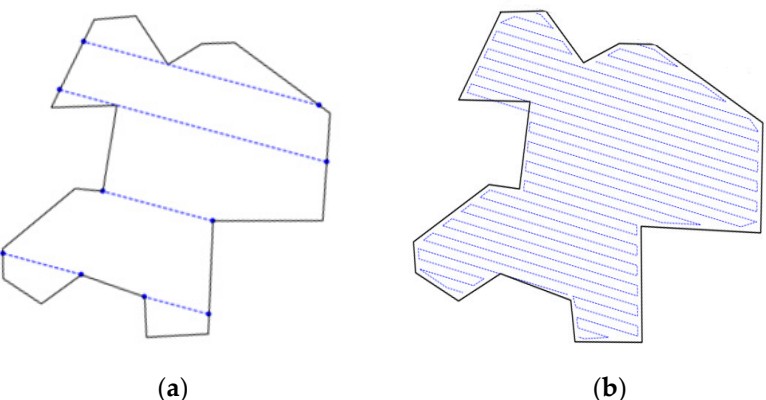

|      (**a**)      |      (**b**)      |

**Figure 13.** The optimal sweep direction and coverage path of drones generated by the BFP method.

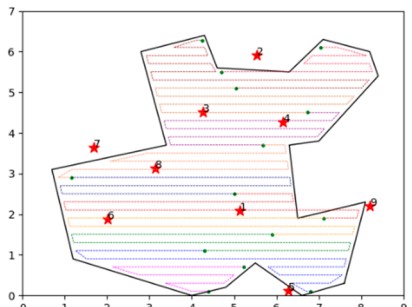

**Figure 14.** The candidate locations of CAs and the segmented scanning trajectories.

### 5.3.2. Results Analysis

The maximum, minimum, and average values of the objective function values (the minimum round-trip distance) and the average calculation time are recorded in Table 5 for each of the four algorithms (GMH, TCH, GMH + LS, and TCH + LS) run 10 times consecutively.

**Table 5.** The results of case experiments.

| **Algorithms** | | **GMH** | **TCH** | **GMH + LS** | **TCH + LS** |
|---|---|---|---|---|---|
| | Best | 41.83 | 38.21 | 37.02 | 37.99 |
| Values of round-trip distance (km) | Worst | 71.59 | 38.21 | 37.99 | 37.99 |
| | Ave | 50.08 | 38.21 | 37.87 | 37.99 |
| Time (s) | | 0.0001 | 0.0003 | 0.044 | 0.055 |
| Improve (%) | | | | 24.38% | 0.58% |

It can be observed that the average objective function value of the feasible solution obtained by the GMH algorithm is 50.08 km, and the average calculation time is about $1.0 \times 10^{-4}$ s; the TCH algorithm obtains a better feasible solution with 38.21 km for all 10 runs, but the average solution time is longer at $3.0 \times 10^{-4}$ s.

Considering the improvement of the feasible solution by the local-search (LS)-based optimization algorithm, the average objective function value of the optimized solution obtained by the GMH + LS algorithm is 37.87 km, with an average improvement of 24.38% compared to the GMH algorithm and an average solution time of 0.044 s. In addition, the optimized solution obtained by the TCH + LS algorithm has an average objective function

value of 37.99 km, is slightly less effective than the GMH + LS algorithm, and has almost no improvement (0.58%) compared to the TCH algorithm. Furthermore, the average running time of the algorithm is 0.055 s and also increases only very slightly.

Figure 15 visualizes and compares the optimal CA deployment and drone scanning mission assignment schemes generated by the four algorithms. And the large red pentagrams and their numerical numbers indicate the locations of the selected CAs. Table 6 shows the four different optimal CAs deployment and scanning mission assignment schemes calculated by the four algorithms. The candidate locations of CAs are numbered from 1 to 9, and the scanning missions are numbered from 1 to 11. The CA column indicates that the location is selected, and the Task column shows the missions assigned to the UVAs owned by these CAs. For instance, the result of the GMH algorithm indicates that locations 1, 4, and 7 are selected to deploy several 3, 4, and 5 CAs and that these CAs need to complete missions numbered (1,3,4), (7,8,9,11), and (2,5,6,10), respectively. From the results, the GMH + LS algorithm and TVH + LS algorithm obtained the same optimal results, in which the deployment locations are 1, 4, and 8, and the scanning missions are (2,3,4,10), (7,8,9,11), and (1,5,6), respectively, with an optimal objective function value of 37.99 km. Similar to the randomly generated experiment results, the TCH algorithm has a similar result compared to the other two algorithms optimized with local search at 38.21 km. It is significantly better than the solution of the GMH algorithm with a more reasonable deployment scheme and smaller objective function values. It once again proves that the TCH algorithm has a significant advantage over the GMH algorithm in finding a better initial solution.

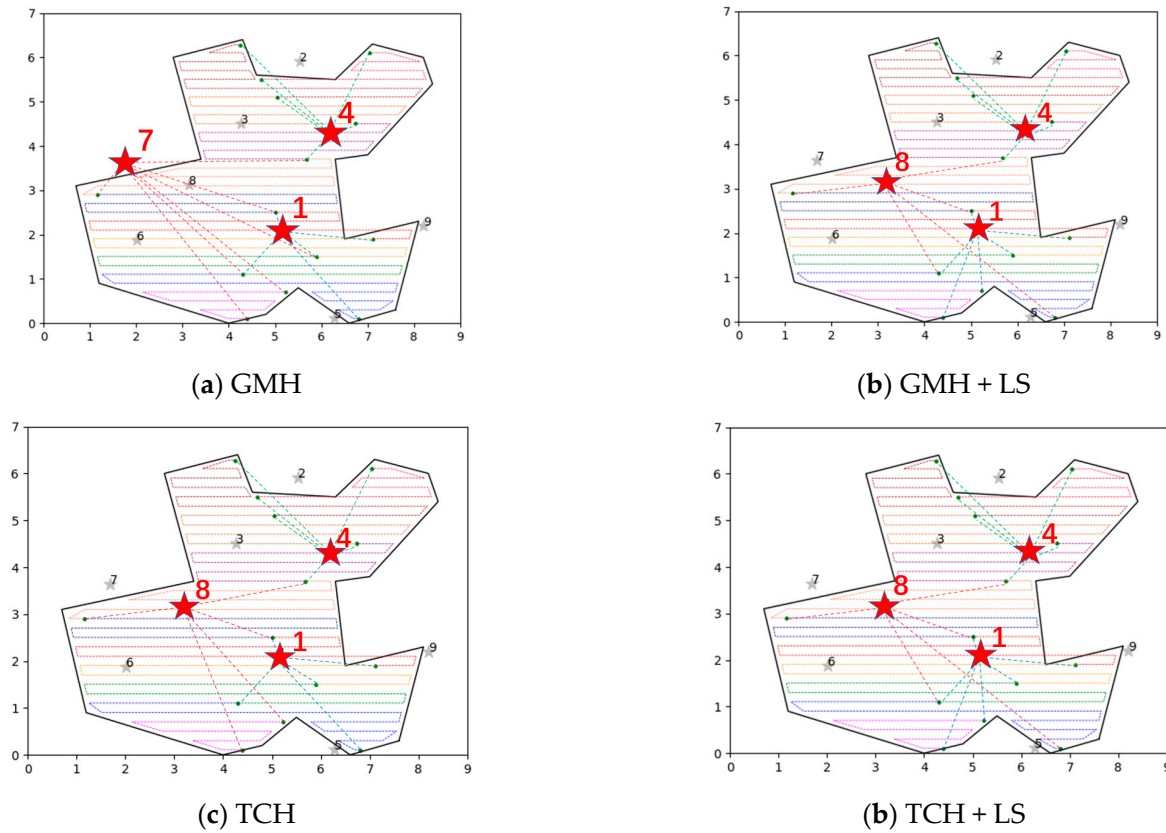

**Figure 15.** Comparison diagram of the four algorithms to generate the optimal solution.

**Table 6.** The CAs deployment and scanning mission assignment scheme.

| GMH | | TCH | | GMH + LS | | TCH + LS | |
|---|---|---|---|---|---|---|---|
| CAs | Missions | CAs | Missions | CAs | Missions | CAs | Missions |
| 1 | 1,3,4 | 1 | 1,2,3,4 | 1 | 2,3,4,10 | 1 | 2,3,4,10 |
| 4 | 7,8,9,11 | 4 | 7,8,9,11 | 4 | 7,8,9,11 | 4 | 7,8,9,11 |
| 7 | 2,5,6,10 | 8 | 5,6,10 | 8 | 1,5,6 | 8 | 1,5,6 |
| Values (km) | 42.5 | Values (km) | 38.21 | Values (km) | 37.99 | Values (km) | 37.99 |

## 6. Conclusions

To overcome the limitations of battery capacity and range of drones, the CA with automatic power generation and autonomous power charging is introduced to provide take-off, landing, and power supply for drones to complete surveillance missions. The problem that integrates the deployment of CAs and drone coverage path planning is investigated and formulated. We propose a solution approach to the CPP problem and a mathematical model for CA deployment and scanning mission assignment to solve this problem. The GMH and TCH heuristic algorithm is designed to obtain the initial solution, and then a local-search-based algorithm is developed to improve it.

Random test experiments with different scales are conducted and indicate the efficiency of the proposed algorithm. Furthermore, we apply the algorithm to solve a practical case. The results show that the overall algorithms can efficiently obtain the optimal solution when dealing with small-scale application scenarios.

The application of drones to undertake operations such as search and surveillance over a wide area has recently attracted a lot of attention. It is a new concept to expand the range of drones by integrating with the CA. There are many interesting research topics worth further exploring, such as the interaction between CAs, the collaboration and relay between drones, and more advanced regional decomposition methods. Due to the complexity of location-routing problems, especially for large-scale problems, more efficient algorithms are valuable for studying in future works.

**Author Contributions:** Conceptualization, W.S. and J.S.; methodology, W.S., Z.L. and J.S.; software, W.S.; data curation, W.S.; writing—original draft preparation, W.S.; writing—review and editing, Z.L. and J.S.; visualization, K.H. All authors have read and agreed to the published version of the manuscript.

**Funding:** This work was supported by the National Natural Science Foundation of China (grant no. 72271241) and the Hunan Key Laboratory of intelligent decision-making technology for emergency management (2020TP1013).

**Data Availability Statement:** No new data were created or analyzed in this study. Data sharing is not applicable to this article.

**Acknowledgments:** They authors thank the editor and anonymous reviewers for their helpful comments on improving the quality of this paper.

**Conflicts of Interest:** The authors declare that they have no known competing financial interests or personal relationships that could have appeared to influence the work reported in this paper.

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
