# Peer review of "Joint Deployment and Coverage Path Planning for Capsule Airports with Multiple Drones"

_drones, doi:10.3390/drones7070457_

Round 1

Reviewer 1 Report

Comments and Suggestions for Authors

It is recommended that the items named Our main contributions to the literature given in the second to last paragraph of the introduction section are presented on separate lines.

It is suggested that the title of Table 1 should be rearranged formally.

Furthermore, it is suggested that the results in Table 3 should be presented more clearly.

Author Response

We very appreciate the valuable comments, and the responses to comments are presented in the attachment. 

Reviewer 2 Report

Comments and Suggestions for Authors

The article presents an original proposal to solve the problem of planning a group of drone missions using airport capsules. The authors proposed an approach to the problem of drone coverage path planning based on the selection of scanning patterns and trapezoidal decomposition. A particularly interesting element of the work is the designed algorithm based on greedy and cluster heuristics. The proposed solution was verified by practical tests and the results are presented in the article.

I have one remark to the authors: Table 1 and Table 3 present the data in an illegible way. Additional lines separating rows and columns would make the data in the table more readable and understandable. It might be a good idea to split Table 3 into two or three separate tables to make it more readable.

Author Response

(The authors gave the same response as above.)

Reviewer 3 Report

Comments and Suggestions for Authors

This paper studies the combination of drones' coverage path planning and the deployment of Capsule Airports.  The authors conducted sufficient comparative experiments. The paper is interesting and well organized. The reviewer has some minor concerns as follows:

(1) The choice of some parameters lacks theoretical basis. For examples:

a) " The side overlap rate refers to the part of overlapping aerial photos of the drone on two adjacent routes, 

which is generally not less than 60%..."

b) ". Generally, the number of candidate locations is 3-6 times the number of desired CAs."

Authors are advised to add references to justify this value choice.

(2) The typesetting of the paper needs to be strengthened. For example, Numbers in Table 3 are indistinguishable.

Comments on the Quality of English Language

None

Author Response

(The authors gave the same response as above.)

Reviewer 4 Report

Comments and Suggestions for Authors

The paper contains a very important discussion and the technical work is sound. The results of the paper is very interesting and the methodology is novel. Only few minor comments as below:

1. Briefly highlight the main advantages of the proposed method than the existing works before the organization of the thesis paragraph.

2. Check your format of first paragraph of Page 8.

3. The Table 3 is poor presented. Please improve.

Comments on the Quality of English Language

Good

Author Response

(The authors gave the same response as above.)
